# The Techniques of Blood Purification in the Treatment of Sepsis and Other Hyperinflammatory Conditions

**DOI:** 10.3390/jcm12051723

**Published:** 2023-02-21

**Authors:** Giorgio Berlot, Ariella Tomasini, Silvia Zanchi, Edoardo Moro

**Affiliations:** 1Department of Anesthesia and Intensive Care, Azienda Sanitaria Universitaria Giuliano Isontina, 34148 Trieste, Italy; 2UCO Anestesia Rianimazione e Terapia Antalgica, Azienda Sanitaria Universitaria Giuliano Isontina, Strada di Fiume 447, 34149 Trieste, Italy

**Keywords:** septic shock, sepsis mediators, hemofiltration, hemoadsorption

## Abstract

Even in the absence of strong indications deriving from clinical studies, the removal of mediators is increasingly used in septic shock and in other clinical conditions characterized by a hyperinflammatory response. Despite the different underlying mechanisms of action, they are collectively indicated as blood purification techniques. Their main categories include blood- and plasma processing procedures, which can run in a stand-alone mode or, more commonly, in association with a renal replacement treatment. The different techniques and principles of function, the clinical evidence derived from multiple clinical investigations, and the possible side effects are reviewed and discussed along with the persisting uncertainties about their precise role in the therapeutic armamentarium of these syndromes.

## 1. Introduction

Sepsis is defined as a life-threatening organ dysfunction caused by a dysregulated host response to infection [1] that is caused by the release of a huge and only partially known number of mediators produced during the interaction between the infecting germ and the patient’s immune system. The possible role of endogenous toxic substances in the pathogenesis of diseases is not a new concept, because since ancient times it was believed that many, if not all, disturbances affecting the humanity were caused by these agents. Consequently, their removal was considered an appropriate therapeutic target; with this aim, bloodletting gained wide popularity, becoming the first procedure of blood purification (BP). However, when in the second half of the 19th century it became clear that an exceedingly high number of microorganisms were responsible for many diseases previously treated with this approach, its use rapidly declined. At present, bloodletting is limited to rather uncommon conditions, including hemochromatosis and polycythemia. The modern era of BP arose in the 1940s, when Kollf et al. started to treat patients with acute or chronic kidney injury (AKI and CKI, respectively) using a cellophane membrane perfused by the patients’ blood to remove uremic toxins [2]. It is remarkable that this approach should be unacceptable in the era of evidence-based medicine (EBM) and Ethical Committees, as the first 16 patients died during the procedures or immediately thereafter and only the 17th patient survived and was discharged home [2]. Some decades later, it appeared that the systemic disturbances associated with severe infections, including fever, arterial hypotension, multiple organ failure, etc., could be ascribed more to the interaction between the host’s immune system and the infecting agent than to the latter only. Moreover, this reaction appeared to be at least partially determined by circulating factors, as the fluid removed from the bloodstream of septic and trauma patients using a cuprophan membrane was able to induce an intense proteolysis in isolated rat limbs, indicating the presence of a filtrable and transmissible factor able to cause the same muscle alterations observed in critical conditions [3]. From then on, an ever-increasing number of substances with both pro- and anti-inflammatory properties produced in these clinical circumstances have been identified [4], and it was hypothesized that their neutralization could positively influence the clinical course of sepsis and septic shock and/or other clinical conditions characterized by an uncontrolled inflammatory reaction. Conversely, in patients with a prolonged length-of-stay (LoS) in the Intensive Care Unit (ICU), these mediators are replaced or, better to say, are counterbalanced by the action of substances with anti-inflammatory capabilities, making them susceptible to infections sustained by low-virulence strains such as *Acinetobacter baumanii*, and to the reactivation of viruses, including Cytomegalovirus and Herpes viruses.

Aiming to neutralize the pro-inflammatory mediators, two different strategies have been developed. The first consists in the administration of inhibitors of a specific mediator or in the blockade of their cellular receptor; however, the results of many randomized controlled trials (RCTs) in ICU patients were largely below the expectations derived from experimental investigations and small Phase I human studies. However, some subgroup analyses indicated an increased survival of patients with elevated blood levels of the specific mediator targeted by the study substance. The use of inhibitors is advocated in the treatment of clinical conditions characterized by their persistent low-level production, including different rheumatologic and chronic inflammatory intestinal diseases.

The second strategy is based on the extracorporeal elimination of germ-derived substances, such as endotoxin or bloodborne mediators produced by the host via different mechanisms, including (a) their convective removal through an artificial membrane used also in Continuous Renal Replacement Treatments (CRRT) whose cutoff value is compatible with their molecular weight (MW); or (b) their adsorption on the membrane surface (Figure 1).

The aims of this review are to describe the principles of the different techniques of BP that are currently used, to evaluate the related clinical experiences, and to illustrate the pros and cons of each in the treatment of septic shock and other similar non-infectious clinical conditions caused by an exaggerated production of hyperinflammatory mediators.

## 2. Rationale of Blood Purification

Different mechanisms have been hypothesized to explain the possible beneficial effects of the elimination of mediators from the bloodstream [5], including:(a)The lowering of both pro- and anti-inflammatory mediators below a threshold level, thus limiting the associated organ damage [6];(b)The passage of mediators from the tissues to the blood and their subsequent extracorporeal clearance along a concentration gradient [7];(c)The restoration of a cytokine gradient between the tissues and the blood, promoting leukocyte chemotaxis [8];(d)The interaction between the membrane and the immune cells, as demonstrated by the modulation of surface molecules during different BP procedures [9].

It is likely that multiple mechanisms (i.e., a + b), maybe in different time windows, cooperate to achieve the therapeutic effect of BP.

## 3. Classification and Principles of Function of the BP Techniques

As stated above, different techniques are used to clear the mediators produced during septic shock or other clinical conditions characterized by elevated levels of inflammatory mediators, such as hemophagocytic syndrome (HS). Their removal is related to the characteristic (a) of the mediators, including their MW and the chemico-physical properties; and (b) of the device used, such as the cutoff value of the membrane, its surface of contact with the substrate to be processed, and the affinity for the substance to be cleared.

Thus, BP can be considered an umbrella term covering different techniques that can be primarily subdivided into blood- and plasma-processing procedures (Table 1). The factors influencing the efficacy of the BP differ according to their principle of function. Consequently, as far as the hemofiltration (HF)-based techniques are concerned, in which the mediators are eliminated by convection, the main determinant of removal is the production of ultrafiltrate (Qf), that, in turn, depends on the blood flowing inside the filter (Qb), the size of the pores, the subsequent sieving coefficient, the surface of the membranes used, and their chemico-physical properties. In contrast, only the Qb accounts for the efficacy of the absorption-based procedures [10]. Despite these differences, both families share a more- or less-pronounced time-dependent decay of the clearance capabilities, and their use can last from 2 to 24 h before the exhaustion of the BP effect.

All BP procedures require a dedicated vascular access using a large-bore catheter and anticoagulation of the extracorporeal circuit using heparin or citrate.

### 3.1. Blood Processing Techniques

#### 3.1.1. Hemofiltration (HF)

HF’s principle of functioning consists in the convective removal of H_2_O and solutes, including mediators, from the bloodstream by means of a synthetic membrane with a cutoff value of ~50–60 kDa, which are used also in CRRTs. The ultrafiltrate (UF) produced has the same electrolyte composition as the plasma. In fact, HF is an umbrella term covering multiple strategies that take advantage of the different amounts of UF considering the therapeutic target (see below). More recently, high-cutoff (HCO) membranes have been developed, but their use is associated with high albumin losses [11]; to overcome this problem, HCO membranes can be used in the diffusive rather than the in convective mode, or by slightly reducing their pore size and surfaces [12]. Independent of the characteristics of the membrane used, the volume of UF produced is related either to the aforementioned variables and to the blood flowing over it per unit time (Qb).

#### 3.1.2. Hemoadsorption (HA)

HA consists in the adhesion of the circulating mediators on the surface of a membrane able to capture them. Four HA techniques have been developed so far [5,10,13]. The first takes advantage of an adsorbing column containing multiple polymixin-immobilized fibers (Toraymixin^®^, Toray Industries, Tokyo, Japan) arrayed into a cartridge to remove the endotoxin molecules from the Qb. Due to this characteristic, its use has been advocated in the treatment of septic shock caused by Gram-negative bacteria only.

The second technique consists in a cartridge containing a synthetic resin constituted by polystyrene and divinylbenzene microbeads (Cytosorb^®^, Cytosorbents Corporation, Monmouth Junction, NJ, USA; Aferetica s.r.l., Bologna, Italy). The wide adsorptive surface (~40.000 m^2^) is able to adsorb hydrophobic pro- and anti-inflammatory mediators with MWs ranging from 5 to 60 kD. Cytosorb^®^ represents an evolution of coupled plasma filtration and adsorption (CPFA; see below), as it uses the same binding resin that is arranged in microtubules instead that in microbeads.

The efficacy of Cytosorb^®^ is concentration-dependent, as substances present in large concentrations are removed more efficiently than those with lower blood levels. Cytosorb^®^ can run in a stand-alone mode or can be associated with a Continuous Renal Replacement Treatment (CRRT) or with an extracorporeal membrane oxygenation (ECMO) apparatus.

The third technique is based on a filter containing a modified AN69 membrane associated with a positively charged polyethyleneimine polymer able to absorb both endotoxin and several different septic mediators (oXiris^®^, Baxter, Meyzieu, France) from the bloodstream, while simultaneously providing CRRT.

The final technique consists in an HA device (Seraph 100^®^, ExThera Medical Corp, Martinez, CA, USA) packed with polymer beads covered with covalent end-point heparin ultra-high-MW polyethylene. This design mimics the heparan sulfate attached on the cell surface, allowing the in vitro binding of toxins, bacteria, and Antithrombin III, thus clearing them from the bloodstream [14]. Due to these properties, the US Food and Drug Administration (FDA) recently approved its use for the treatment of COVID-19 patients.

### 3.2. Plasma Processing Techniques

Three techniques are currently used. They include:Plasmapheresis (PF), which is based on the selective removal of one or more plasma components (lipoproteins, paraproteins, etc.), and is not currently used in the treatment of septic shock;Plasma exchange (PEX), consisting in the removal of one or more volumes of plasma, which is replaced with donors’ plasma or albumin. The rationale of PEX consists in the removal of “toxic substances” and the supply of a large amount of plasma components whose absence is considered responsible for the disorder (i.e., ADAMTS 13 for patients with thrombotic thrombocytopenic purpura [15]). Ideally, the best candidate substance for removal by PEX should have a high MW, small volume of distribution, long half-life, and low turnover rate [15];Coupled plasma filtration and adsorption (CPFA), which basically consists in a three-step process: (1) the partial extraction of plasma from the blood via a plasma filter; (2) its processing within a cartridge, where a number of mediators are absorbed by a synthetic resin arranged in microtubules; and (3) reinfusion of the purified plasma upstream of a second filter used for continuous veno-venous hemodiafiltration in cases of concomitant AKI. The adsorptive capabilities of the resin are exhausted after 10 h, but the CRRT can continue beyond this limit by excluding the plasma processing unit.

## 4. Clinical Research Evidence

### 4.1. Hemofiltration

As many clinical investigations have involved only a relatively limited number of patients or are retrospective, the result of prospective RCTs are reported in Table 2 (after).

On the basis of previous investigations, which demonstrated a dose–effect relationship between the UF and survival [16], or reduced need for a vasopressor [17], it has been hypothesized that very elevated UF values per unit time (Qf) could be associated with an improved survival of septic shock patients treated with HF. Indeed, despite some studies demonstrating encouraging results [18], a large RCT using high-volume HF (HVHF) that compared an elevated (70 mL/kg/h) with a normal (35 mL/kg/h) Qf in 137 septic shock patients failed to confirm these findings [19] and this approach has been largely abandoned. Furthermore, the higher Qf reportedly determined a significant loss of antibiotics [20]. An evolution of this technique is the cascade HVHF, which was developed to selectively remove medium molecular weight (MW) molecules while retaining those with lower MW, including vitamins, nutrients, and drugs. The technique combines two different hemofilters with different cutoff values: the first hemofilter, with a larger cut-off, produces an ultrafiltrate containing both large and small MW molecules and flows though another one with a smaller cutoff; then, only medium-MW molecules will be cleared and those with lower MW are reinfused back to the patient as predilution fluid before the first hemofilter [21]. Despite the result of an experimental study that demonstrated a reduction in the need for a vasopressor in a porcine model of sepsis, a recent study of cascade HVHF failed to demonstrate any beneficial effect when compared with standard care in septic shock patients [21,22]. The use of HCO membranes has been associated with the reduction of several inflammatory mediators in some studies [12,23], but another investigation did not confirm these findings [24].

### 4.2. Hemoperfusion

#### 4.2.1. Endotoxin Adsorption

Whereas this approach is commonly used in Japan on the basis of clinical investigations and clinical registries, in Western countries different RCTs aiming to assess the efficacy of this procedure produced conflicting results. In the Early Use of PolymixinB Hemoperfusion in Abdominal Septic Shock (EUPHAS) study [25], patients treated with this technique demonstrated hemodynamic and respiratory improvements associated with a trend toward a better outcome. However, a subsequent study performed in patients with septic shock due to peritonitis, the ABDOMIX Trial [26], demonstrated a trend of increased mortality in the treatment group. Finally, the Evaluating the Use of Polymixin B Hemoperfusion in the Randomized Controlled of Adults Treated for Endotoxemia and Septic Shock Trial (EUPHRATES) performed in septic shock patients with elevated blood endotoxin levels measured with the Endotoxin Activity Assay (EAA) demonstrated a beneficial effect on different variables, including survival, only in patients with high EAA results [27]. Taken together, it appears that this approach could be effective when the mortality of the control group ranges from 30 to 40%, and/or with elevated blood endotoxin levels. It is also possible that the somewhat divergent findings between Japanese and Western RCTs could be ascribed to different genetic and enzymatic profiles.

#### 4.2.2. Cytosorb^®^

Although some experimental and clinical investigations demonstrated that the use of Cytosorb^®^ is associated both with the reduction of blood levels of many inflammatory cytokines, with the reduction of the vasopressors and with the improved survival of patients with septic shock [28,29,30,31], other studies failed to confirm these findings [32,33,34]. Recently, Hawchar et al. [35], evaluating 1434 patients with different clinical conditions including 936 cases of septic shock treated with Cytosorb^®^ demonstrated that, although the primary outcome of hospital mortality was higher than that reported in other studies (59% vs. 46.5%, respectively), it was lower than expected according to the APACHE II score (66%). While it is difficult to draw a definite conclusion, it is possible that different variables can account for these contrasting results, including the heterogeneity of patients treated, the intensity of the treatment, and the timeframe of initiation with the clinical course. In fact, in a group of septic shock patients, Berlot et al. demonstrated that in survivors either the amount of blood processed was higher or the interval of time elapsing from the onset of shock and the start of Cytosorb^®^ was shorter than in nonsurvivors [36]. To maximize the effect of Cytosorb^®^, Bottari et al. [37] advocated the replacement of the cartridge every 12 instead of every 24 h, at least in the initial phase of the treatment, to take full advantage of the adsorptive capabilities of the resin.

#### 4.2.3. oXiris^®^

Experimentally, this technique was demonstrated to have the same endotoxin-removing capabilities as Toraymixin^®^, and was similar to Cytosorb^®^ regarding the clearance of mediators. Currently, the clinical experience is limited and basically consists in small case series of patients with septic shock and/or COVID-19, in whom improvements of the hemodynamic conditions, decreases in the blood concentrations of endotoxin and septic mediators, and the decrease of expected mortality was observed [38,39,40]. However, as stated by Li et al. [41], not dissimilar to what is stated above, the heterogeneity of the patients and the different underlying conditions create background noise and prevent the establishment of the real role of this procedure.

#### 4.2.4. Seraph 100^®^

In the absence of clinical trials, the role of this device is still uncertain. Recently, Eden et al. [14] demonstrated a rapid resolution of bacteremia in a group of CKI patients undergoing RRT.

## 5. Plasma Exchange

If the roles of the different HA techniques of HA are not yet clear, even less definite is that of PEX. Besides the time-honored indications in critically ill patients [15], use of PEX in septic shock patients appears somewhat overshadowed by HA. This notwithstanding, a recent RCT involving 40 patients [42] demonstrated a trend for better survival and the improvement of multiple organ failure in patients treated with a single PEX with an exchange volume of >3000 mL of plasma associated with standard treatment ST as compared with the control group which received the ST only; as might be expected, the decrease of sepsis biomarkers and the replenishment of factors supplied with the plasma, including Protein C, Protein S, and ADAMTS 13, were observed in the PEX group, but not in the control group. Moreover, patients in the PEX group were weaned faster from the vasopressors and had a more pronounced decrease of blood lactate levels; similar results were demonstrated by David et al. [43], who observed a decreased need for vasopressors in a group of septic shock patients.

## 6. Coupled Plasma Filtration and Adsorption

Different investigators reported either the improvement of hemodynamic conditions or better outcomes with CPFA in several relatively small case series of septic shock patients [44,45,46,47,48,49,50]. To elucidate this potential role of CPFA in septic shock patients, an initial RCT (COMPACT 1) involving 192 out of 330 pre-planned patients was launched. The trial was suspended when an interim evaluation failed to show any survival benefit in the treatment group; yet, a post hoc analysis demonstrated that survivors had a larger volume of plasma processed (Vp) (≥0.20 L/kg/session) than controls [47]. To evaluate this dose–effect relationship, a second RCT was subsequently started (COMPACT 2) using this value as the threshold Vp for the treatment group. However, this second RCT was also prematurely stopped when an intermediate analysis RCT involving 115 patients demonstrated an increased mortality in patients treated with CPFA [51]. Similar results have been reported by Gimenez-Esparta in another RCT (ROMPA) in 49 septic shock patients treated with CPFA [52]; however, this study was underpowered to draw definite conclusions.

Overall, these results caused the virtual disappearance of CPFA from the therapeutic armamentarium used in critically ill patients. Indeed, the puzzling is question is: why did the RCTs about the use of the CPFA failed to demonstrate any beneficial effect, whereas in single-center studies the outcome was positively influenced by this technique? In other words, has the jury reached the right verdict? As an example, Mariano et al. [53] demonstrated that in a group of severely burned and AKI patients, those treated with the CPFA (n: 39) had a significantly better outcome as compared with those of the control group (n: 87) who were treated with the RRT only (survival rate 51.1% vs. 87.1, respectively, *p* < 0.05). It is conceivable that patients treated in a single ICU take the maximal advantage from the experience of the local ICU staff, while the results of RCTs can be influenced by the co-existence of ICUs with different volumes of procedures.

## 7. Discussion

Despite several years of use and thousands of patients enrolled in clinical trials with different BP techniques, a number of grey areas persist. The most recent guidelines of the Surviving Sepsis Campaign do not advise for or against leaving centers free to adopt their own policy of BP [54]. Moreover, due to the methodological biases encountered in different investigations, some authors advocate multi-center RCTs that fully satisfy the EBM criteria [55,56]; yet, these studies appear difficult to launch due to the widespread use of these procedures, which makes the implementation of a clinical trial sponsored by the manufacturers or by health authorities unlikely.

The uncertainties concerning the use of BP in septic shock patients and/or in those with severe hyperinflammatory disease are caused by several factors other than infections. These factors can be summarized as follows:

The selection of patients. According to multiple studies, the best candidates are patients with septic shock whose source of sepsis has been identified and/or surgically treated. Due to their costs and inherent risk of iatrogenic complications, the risk/benefit ratio should be considered in every BP candidate.

The timing of initiation. As has been demonstrated with antibiotics, it appears that the early initiation of BP in the hyperinflammatory phase of septic shock is associated with a better outcome. Even in the absence of specific studies, it is reasonable that the same consideration applies in hyperinflammatory conditions other than septic shock [36]. However, there is a lack of clarity regarding their possible role in chronic critically ill patients in whom anti-inflammatory mediators prevail and set the stage for infections with opportunistic germs and viral reactivation.

The intensity of the procedure. It appears that a dose–effect relationship exists for BP. However, the risk of elimination of drugs and nutrients should not be overlooked, especially in the presence of elevated values of Qb or Qf [57]. A U-shaped curve can be hypothesized, in which undesirable effects, such the low removal of mediators and the clearance of antibiotic, are located at the opposed extremities, whereas the beneficial effects lay somewhere in between (Figure 2). As this point is difficult to establish, repeated measurements of the blood concentrations of antibiotics and other drugs are warranted, especially in the initial phase of a BP procedure, when the clearance capabilities are maximal and can impede the rapid achievement of an effective plasma concentration, which is a therapeutic target of pivotal relevance.

d.The assessment of the efficacy. The outcome of septic shock patients and of patients with non-septic hyperinflammatory conditions can be influenced by factors other than the BP used, including the appropriateness of the antibiotic treatment, the timely and complete drainage of septic foci, underlying conditions, etc. Thus, survival by itself does not represent a reliable marker of the efficacy of BP; consequently, other biological and clinical variables, such as the variation of the blood lactate levels and the changes in the need for vasopressors, can be used as proxies of efficacy [5].e.The choice of the clinical situation. As stated above, patients with a prolonged LoS in ICU can undergo a biphasic clinical course, the first being characterized by a hyperinflammatory reaction that can be followed by a second one associated with the reduction of the immune capabilities caused by the production of substances with anti-inflammatory properties. These patients are usually old and with several frailties associated with pre-existing irreversible chronic conditions, such as chronic heart failure and obstructive pulmonary disease and worsening of chronic kidney disease. These patients often survive the disease that prompted the ICU admission, but their subsequent clinical course is marked by the occurrence of a number of different complications that make their survival unlikely, including malnutrition, difficulty in weaning from mechanical ventilation, skin ulcers, reinfections, etc. The possible role, if any, of BP in these chronic critically ill patients is not yet clear since most clinical investigations concerning BP treated patients in the initial hyperinflammatory phase and not in the second stage of the disease.f.Undesired effects other than drug removal. In addition to the iatrogenic risks associated with indwelling large-bore catheters and the need of anticoagulation, all BP procedures can induce an undesired hypothermia due to the extracorporeal circuitry; to overcome this effect, all the currently used devices can warm the blood of the re-entry segment.g.Lack of precision. BP techniques efficiently clear from the bloodstream all substances with certain chemico-physical properties, independent from their role in that timeframe. In many cases, the rule of “one size fits all” was and is still the rule for BP, and for many other treatments currently used in critically ill septic patients [58]. This is far removed from precision medicine in which the treatment is tailored to the needs of the individual patient. However, this approach is still experimental in critically ill septic patients [59].

## 8. Conclusions

Multiple factors, including the advancing age of the general population, the widespread use of invasive procedures, the use of immunodepressant drugs, and the ever-increasing occurrence of antibiotic resistance, make it so that the occurrence of septic shock is, and will remain in the future, a highly relevant issue among critically ill patients. Presently, the administration of appropriate antibiotics represents the only undiscussed therapeutic option for its causal treatment. Despite the somewhat conflicting results of many RCTs, BP techniques can be a valid adjunctive measure for these patients, provided that they are applied appropriately and considering their potential scavenging effects on antibiotics and other therapeutic agents.

## Figures and Tables

**Figure 1 jcm-12-01723-f001:**
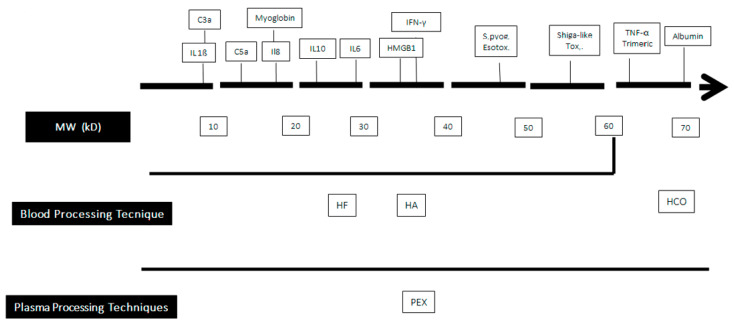
Clearance capabilities of blood purification techniques according to the molecular weight of some bloodborne substances. MW: Molecular Weight; HF: Hemofiltration; HA: hemadsorption; PEX: plasma exchange; HCO: high cut-off membrance.C3a: Activated Complement factor 3; IL: interleukin; HMG 1: High Mobility Group Box 1; IFN-γ: γ Interferon.

**Figure 2 jcm-12-01723-f002:**
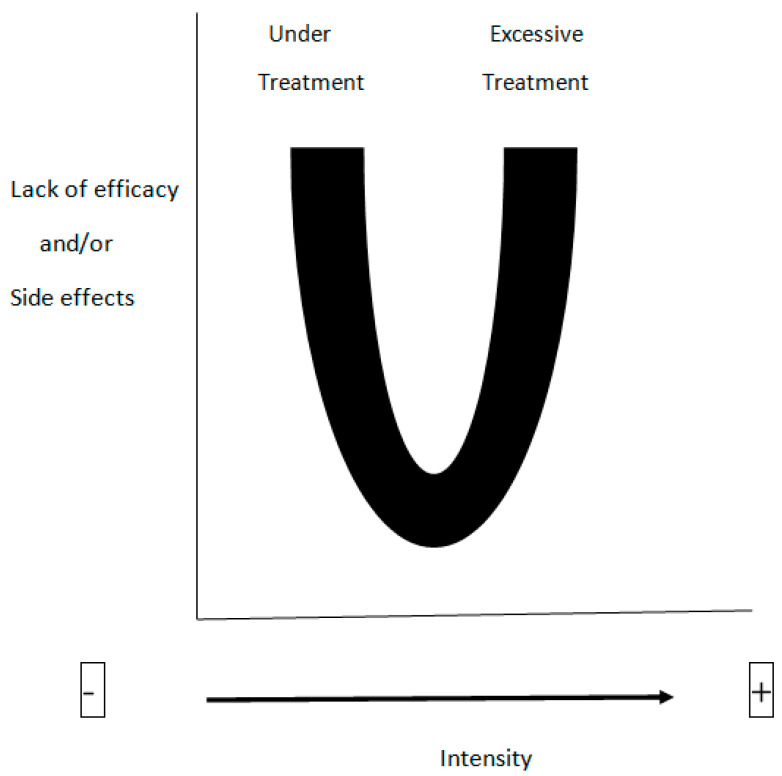
Hypothetical dose–effect relationship between the intensity of the treatment and the occurrence of side effects.

**Table 1 jcm-12-01723-t001:** Techniques used in BP. Legend: CPFA^®^: coupled plasma filtration and adsorption.

Substrate	Technique/Brand	Mechanism
Blood	Ultrafiltration	High-volume ultrafiltration
		High-cutoff membrane
	Hemoadsorption	Toraymixin^®^
		oXyris^®^
		Cytosorb^®^
		Seraph^®^
Plasma	Plasma exchange	
	Ultrafiltration + plasma adsorption	CPFA^®^

**Table 2 jcm-12-01723-t002:** Results of some RCTs of BP in septic shock patients.

Study/Author	BP	Treatment Group N.	Control Group N.	Results
IVOIRE	HVHF	66	71	No difference in hospital mortality
EUPHAS	Toraymixin^®^	30	34	Improved hemodynamics and survival in the treatment group
ABDOMIX	Toraymixin^®^	119	113	Non-significant increase in mortality and no improvement in organ failure in the treatment group
EUPHRATES	Toraymixin^®^	84	78	Toraymixin^®^ compared with sham treatment did not reduce mortality at 28 days
Supady et al. *	Cytosorb^®^	17	17	Excess mortality in the treatment group
ROMPA	CPFA	19	30	No difference in hospital mortality
COMPACT 1	CPFA	91	93	No difference in hospital mortality
COMPACT 2	CPFA	63	52	Excess mortality in the treatment group

* In COVID-19 patients in association with ECMO.

## Data Availability

Not applicable.

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
