# Peer review of "The Techniques of Blood Purification in the Treatment of Sepsis and Other Hyperinflammatory Conditions"

_jcm, 2023, doi:10.3390/jcm12051723_

Round 1

Reviewer 1 Report

Berlot G et al reviewed the current clinical applications of blood purification therapies (BP) during the treatment of septic shock. In this narrative review, they discuss the rationale of blood purification and the available clinical experiences. They reported the impact of specifically developed membranes with large pores or absorption properties, hemoperfusion, and sorbent cartridges or columns, such as high cut-off membranes, the Oxiris® -AN69 membrane, CytoSorb®, and Seraph 100, and plasmapheresis and plasma exchange.

They concluded that all these techniques have been evaluated in small studies series or are under evaluation for improving patient outcomes in septic shock. Therefore, they summarize the few consolidated data and uncertainties concerning the use of these blood purification therapies.

General comment

The Authors faced a debated issue, and the message coming from this narrative review is interesting. To my mind, the paper has some critical points.

Major points:

1)      The first critical point is that the reported experiences are incomplete. The Authors dealt with high HV-hemofiltration, high cut-off membranes, hemoperfusion (Polymixin B cartridge, Cytosorb, oXiris, and Seraph 100), and plasma exchange. They did not report the experience done with coupled plasma filtration and adsorption in septic shock patients with associated AKI by many groups, including the present Authors (for details, see as main references: 1) Ronco C, et al. Crit Care Med 2002; 30:1250-5;  2) Formica M et al. Intensive Care Med 2003; 29:703-8;   3) Mariano F et al. Blood Purif 2004; 22:313-9;   4) Livigni S et al. BMJ Open 2014; Jan 8; 4(1):e003536;  5) Berlot G, et al Blood Purif. 2018;46(4):274-278.  6) Garbero E et al. Intensive Care Med. 2021 Nov;47(11):1303-1311.    6) Mariano F et al. Burns 2020;46:190-8. Please, in a separate subchapter discuss and comment on these papers and similar others.

2)      I appreciate the points stated by the Authors in the conclusions, and I ‘m aware that the Compact 2 study was prematurely interrupted for an excess of mortality in CPFA treated group. However, many Authors pointed out the intrinsic limits of larger randomized trials in studying the efficacy and the improved outcome of these new specific therapies, mainly when sophisticated and complicate methodologies were applied (for an extensive discussion of the issue, please see Vincent JL. Minerva Anestesiol 2015; 81:122–4). We are moving towards precision medicine, which needs a well-defined target population. I think these considerations could be important, and they deserve to be added to your clear and thoughtful reflections.

Minor points

1) In Table 1 last line, please change  “Ultrafiltration+hemoadsorption” in “Ultrafiltration+plasma adsorption”

Author Response

Trieste 8.1.2023

Dear Rewiever # 1,

We made all the suggested modifications and corrections.

In detail:

  • Major points
  1. The CPFA issue is expanded and the technique and the related clinical investigations are discussed in detail.
  2. The issue of the precision medicine is discussed in the final part of the revised manuscript.

  • Minor points
  1. Table 1 has been modified according to then indications.

All changes are marked in yellow

On behalf of all other Authors,

Prof. Giorgio Berlot

UCO Anestesia Rianimazione e Terapia Antalgica

Strada di Fiume 447 34149 Trieste

Tel 04039904540

Fax 040912278

Mail: [email protected]

Reviewer 2 Report

Dear authors,

Your manuscript is a narrative mini-review on blood purification techniques summarizing clinical evidence regarding their role in the treatment of sepsis and septic shock. It is comprehensible, concise and informative to the reader.

I have a few comments that you may find helpful in improving your work

1. Your review is very brief regarding the Clinical data. You may consider reporting in more detail the major clinical trials i.e.  the type of study (retrospective or prospective), the number of patients studied and the severity of sepsis etc.

2. You may list the relevant clinical studies in a Table depicting main characteristics of study population, method of BP used and main results.

3. A Discussion section is required to provide a critical overview of your findings. In the Discussion you should include the controversies regarding blood purification and comment in detail on the methodological bias of the studies (inhomogeneous study population, variation in timing, intensity and duration of BP etc). Also include the list of uncertainties in the last paragraph of your Conclusions by numbering them. You may also add a short comment on the limitations of your review. You may close by giving your personal view of the future perspectives of the method.

4. Please keep the Conclusion short and comprehensive. The text in your Conclusion section should be included in the Discussion. For your conclusion add a short text (4-6 rows) to give the overview of where BP stands on sepsis treatment today and what to expect in the future.

Finally, the English language needs editing. Also certain expressions need to be corrected. For example:

“up-to-date informations” should be “recent evidence”

“organ failures” should be “multiple organ failure”

 “request of catecholamines” should be “vasopressor needs”

“the choice of patients” should be “selection of patients”

“middle molecules” should be “ medium molecular weight molecules”

“released its use” should be “ approved”

“clinical experiences” should be “clinical data” or “clinical evidence”

“sepsis mediators” should be “inflammatory mediators”

“gram-germs” should be “gram negative bacteria”

“functioning” should be “function” or “operation”

Author Response

Dear Rewiever # 2,

We made all the indicated modifications.

In detail:

  • Major points
  1. The section of clinical investigations has been expanded and a specific table has been added
  2. A Discussion section has been added
  3. A short Conclusion section has been added
  4. All the editing suggestions have been followed
  5. 2 explanatory figures have been added.

All changes are marked in yellow.

On behalf of all other Authors,

Prof. Giorgio Berlot

UCO Anestesia Rianimazione e Terapia Antalgica

Strada di Fiume 447 34149 Trieste

Tel 04039904540

Fax 040912278

Mail: [email protected]

Round 2

Reviewer 1 Report

The paper is significantly improved, however, some crucial points remain in the chapter regarding the CPFA. Reporting the clinical experience on CPFA, the authors cited several reports demonstrating the improvement of hemodynamic parameters involving few patient cases. As a clinical trial, they cited the Compact 1 and Compact 2, and the ROMPA study, this latter involving 49 patients (19 treated by CPFA and 30 controls). All these trials failed in demonstrating an improved survival, like all other sophisticated BP techniques applied in the past years.

About CPFA, the Authors did not mention the large experience of CPFA in septic shock-burned patients with AKI needing KRT (Burns 2020;46:190-8). In this paper, the 39 severely burned patients with septic shock  treated with CPFA as an adjunctive treatment to CKRT was compared with the control groupby of 87 burn patients on CKRT with the same baseline characteristics. Even if this study was not randomized, it had the great advantage of having been done in a peculiar patients’ population, with a well-defined septic process, and expected mortality rate. This study, done by nephrologists and intensivists, documented an improved mortality rate for burn CPFA-treated patients. As for other BP techniques in septic shock patients, this study posed the question of the role of sophisticated and complicated methodologies in a specific population, and of the need for precision medicine (for an extensive discussion of the this general issue, please see Vincent JL. Minerva Anestesiol 2015; 81:122–4).

Author Response

Trieste 13.2.2023

Rewiever # 1,

We made all the indicated modifications.

In detail:

  1. The section of CPFA has been expanded by adding a reference about burn and AKI patients and comparing the results of RCT vs single-center clinical investigations
  2. The issue of precision medicine has been expanded

On behalf of all other Authors,

Prof. Giorgio Berlot

UCO Anestesia Rianimazione e Terapia Antalgica

Strada di Fiume 447 34149 Trieste

Tel 04039904540

Fax 040912278

Mail: [email protected]

Reviewer 2 Report

Dear authors,

Thank you for the revised manuscript. You have followed my suggestions and revised your manuscript substantially. 

Author Response

Trieste 13.2.2023

Rewiever # 2,

Thank you for your valuable remarks that substantially improved the manuscript.

On behalf of all other Authors,

Prof. Giorgio Berlot

UCO Anestesia Rianimazione e Terapia Antalgica

Strada di Fiume 447 34149 Trieste

Tel 04039904540

Fax 040912278

Mail: [email protected]